# Direct Lysis RT-qPCR of SARS-CoV-2 in Cell Culture Supernatant Allows for Fast and Accurate Quantification

**DOI:** 10.3390/v14030508

**Published:** 2022-02-28

**Authors:** Nicky Craig, Sarah L. Fletcher, Alison Daniels, Caitlin Newman, Marie O’Shea, Wenfang Spring Tan, Amanda Warr, Christine Tait-Burkard

**Affiliations:** 1The Roslin Institute and Royal (Dick) School of Veterinary Studies, University of Edinburgh, Easter Bush, Midlothian EH25 9RG, UK; nicky.craig@roslin.ed.ac.uk (N.C.); sarah.fletcher@roslin.ed.ac.uk (S.L.F.); a.daniels@sms.ed.ac.uk (A.D.); caitlin.newman@roslin.ed.ac.uk (C.N.); marie.oshea@roslin.ed.ac.uk (M.O.); wenfang.spring.tan@ed.ac.uk (W.S.T.); amanda.warr@roslin.ed.ac.uk (A.W.); 2Division of Infection Medicine, University of Edinburgh, Edinburgh EH16 4SB, UK

**Keywords:** SARS-CoV-2, real-time PCR, COVID-19, direct lysis, diagnostics, coronavirus, nidovirus, enveloped viruses, BHV, IAV, RSV

## Abstract

Studying the entire virus replication cycle of SARS-CoV-2 is essential to identify the host factors involved and treatments to combat infection. Quantification of released virions often requires lengthy procedures, whereas quantification of viral RNA in supernatant is faster and applicable to clinical isolates. Viral RNA purification is expensive in terms of time and resources, and is often unsuitable for high-throughput screening. Direct lysis protocols were explored for patient swab samples, but the lack of virus inactivation, cost, sensitivity, and accuracy is hampering their application and usefulness for in vitro studies. Here, we show a highly sensitive, accurate, fast, and cheap direct lysis RT-qPCR method for quantification of SARS-CoV-2 in culture supernatant. This method inactivates the virus and permits detection limits of 0.043 TCID_50_ virus and <1.89 copy RNA template per reaction. Comparing direct lysis with RNA extraction, a mean difference of +0.69 ± 0.56 cycles was observed. Application of the method to established qPCR methods for RSV (-ve RNA), IAV (segmented -ve RNA), and BHV (dsDNA) showed wider applicability to other enveloped viruses, whereby IAV showed poorer sensitivity. This shows that accurate quantification of SARS-CoV-2 and other enveloped viruses can be achieved using direct lysis protocols, facilitating a wide range of high- and low-throughput applications.

## 1. Introduction

Severe acute respiratory syndrome coronavirus 2 (SARS-CoV-2) is the causative agent of coronavirus disease 2019 (COVID-19), which emerged towards the end of 2019 [1]. From the original outbreak in China, the virus spread rapidly across the globe, leading the World Health Organization to assign the virus “pandemic status” in March 2020. Since then, millions of confirmed cases and deaths have been associated with SARS-CoV-2 infection.

SARS-CoV-2 is an enveloped, positive-sense RNA virus, belonging to the family *Coronaviridae* in the order *Nidovirales*. Coronaviruses (CoVs) are capable of infecting a wide variety of mammalian and avian species. In most cases, they cause respiratory and/or intestinal tract disease. Human coronaviruses (hCoVs) are known as major causes of the common cold (e.g., HCoV-229E and HCoV-OC43). However, the emergence of new hCoVs of zoonotic origin has shown the potential of CoVs to cause life-threatening disease in humans, as was demonstrated during the 2002/2003 SARS-CoV-1 epidemics, the ongoing MERS-CoV epidemics in the Middle East, and now the SARS-CoV-2 pandemic [2,3].

The global vaccination effort has eased the burden of COVID-19 slightly, but there remains an urgent need for effective antiviral treatments, especially for early administration, outpatient treatments to prevent progression to severe disease, particularly in high-risk patients. Early efforts to identify interventions inhibiting SARS-CoV-2 replication relied on the repurposing of existing, approved drugs with known toxicity profiles rather than de novo drug development. Whilst hundreds of drugs were trialed in hundreds of thousands of patients, only seven drugs were given a grade A by the CORONA Project database in treatment efficacy and/or research prioritization for outpatient treatments: Bamlanivimab + Etesevimab and Sotrovimab, Budesonide, and Remdesivir (all A for both), and Ciclesonide, Fluvoxamine, and Molnupiravir (A for research prioritization only) (https://cdcn.org/corona/, accessed on 16 February 2022). It is therefore essential to continue the effort to find new and repurposed treatments against SARS-CoV-2 infection. Examining the whole replication cycle, including virus release, will reveal more candidates for further investigation and narrow the search to drugs that truly reduce the viral load.

In both diagnostic RT-qPCR and in vitro studies of SARS-CoV-2, the requirement to extract and purify viral RNA (vRNA) prior to measuring virus RNA copy numbers is expensive in terms of both time and resources. Methods for lysis and direct RT-qPCR of viral samples without RNA purification have previously been developed for the study of influenza (cell lysate, [4]), Dengue virus (cell supernatant [5]), Zika virus (patient samples [6]), norovirus and hepatitis A Virus (foods [7]). In some of these, the use of expensive, commercially available lysis buffers limits applicability for high-throughput screening, and none of the publications analyzed the impact of lysis buffer on efficiency and sensitivity of the assays. For SARS-CoV-2, several groups have developed direct RT-qPCR methods for detection aimed at patient swabs. The most commonly used method is direct use of swabs following a heating step: 30 min at 65 °C or increasingly shorter periods up to 95 °C. With or without addition of commercial buffers or detergents, a Ct difference between four and seven cycles was observed compared with extracted vRNA [8,9,10,11,12,13,14,15,16]. Other methods include the addition of proteinase K to patient swab samples, showing 4–6 cycle differences, but no proof of virus inactivation was shown for these samples [17,18]. Commercial kits or homemade detergent-based kits showed good correlation with positive clinical samples, but virus inactivation and loss of Ct were not determined [19,20,21]. The reduction in sensitivity, lack of inactivation proof, or the reliance on expensive proprietary lysis buffers make many of these methods unsuitable for quantification in in vitro-amplified viral culture supernatants.

Here, we show a method for direct lysis and RT-qPCR of vRNA in culture supernatant using a cheap, non-commercial IGEPAL CA-630 (IGEPAL-630)-based lysis buffer, which completely inactivates SARS-CoV-2 (>1E6 TCID_50_/mL reduction). The assay shows high sensitivity, detecting <0.0043 TCID_50_ per reaction in lysate and to <1.89 copy per reaction using RNA template-spiked mock lysate. The method described here can be used to accurately, rapidly and cost-effectively quantify SARS-CoV-2 production in cell culture supernatant, allowing for faster workflows, saving time and resources in routine virological applications as well as high-throughput screening.

## 2. Materials and Methods

### 2.1. Resource Availability

Further information and requests for resources and reagents should be directed to and will be fulfilled by the lead contact, Christine Tait-Burkard (christine.burkard@roslin.ed.ac.uk)

### 2.2. Cells and Viruses

Vero E6 (ATCC CRL-1586) cells were maintained as monolayer cultures in Dulbecco’s modified Eagle medium (DMEM, Sigma, Hilden, Germany), supplemented with 10% heat inactivated Fetal Bovine Serum (Gibco, Life Technologies Ltd., Paisley, UK), 1× Ultraglutamine-I (Lonza, Verviers, Belgium), 1× Non-essential Amino Acids (NEAA, Lonza), and 1× Penicillin-Streptomycin (Gibco, Life Technologies Ltd., Paisley, UK) (complete DMEM) at 37 °C in a 5% CO_2_ atmosphere.

Samples from confirmed COVID-19 patients were collected by a trained healthcare professional using combined nose-and-throat swabbing. The sample was stored in virus transport medium prior to cultivation and isolation on Vero E6 (ATCC CRL-1586) cells following sterile filtration through a 0.1 µm filter. Samples were obtained anonymized by coding, compliant with Tissue Governance for the South East Scotland Scottish 279 Academic Health Sciences Collaboration Human Annotated BioResource (reference no. SR1452). Virus sequence was confirmed by Nanopore sequencing according to the ARCTIC network protocol (https://artic.network/ncov-2019, accessed on 7 November 2021), amplicon set V3, and validated against the patient isolate sequence. The main virus isolate used in this project was EDB-2 (Scotland/EDB1827/2020, UK lineage 109, B1.5 at the time, now B.1). Methods were also confirmed using EDB-1, EDB-B117-2 (alpha variant), and EDB-B16172-1 (delta variant).

Bovine herpes virus strain BHV-1-GFP-VP26 was produced on MDBK cells (ECACC 90050801) [22]. Human H1N1 influenza A virus strain A/PR/8/34 (PR8) was the cell-adapted variant of the NIBSC vaccine strain produced on MDCK cells (ECACC 84121903) [23]. Respiratory syncytial virus strain RSV-BT2a, subgroup A patient isolate, was produced on HEp-2 cells (ECACC 86030501) [24].

### 2.3. Primer Optimization Using In Vitro Transcribed RNA Fragments

Fragments of the SARS-CoV-2 reference sequence Wuhan-Hu-1 NC_045512.2 were purchased as double-stranded DNA fragments. The fragments do not correspond to full genes. Fragment 10 (IDT, Leuven, Belgium), containing parts of RdRp (nsp12, orf1ab, bp15,421–16,670), and fragments 18 (IDT, bp27,627–29,126) and 19 (Life Technologies, Thermo Fisher, Regensburg, Germany, bp29,109–29,590), containing parts of N (orf10), were cloned into the pCR-Blunt-II vector prior to in vitro transcription using the mMessage mMachine SP6 (Invitrogen, Thermo Fisher, Vilnius, Lithuania) transcription kit, according to the manufacturer’s instructions. In vitro transcription of the templates results in the following size fragments: 10–1476 bp, 18–1726 bp, and 19–725 bp.

To test primer efficiency, template RNA (tempRNA) 10 was used for DZIF RdRp (F-GTGARATGGTCATGTGTGGCGG, R-CARATGTTAAASACACTATTAGCATA), tempRNA 18 for CDC N1 (F-GACCCCAAAATCAGCGAAAT, R-TCTGGTTACTGCCAGTTGAATCTG) and N3 (F-GGGAGCCTTGAATACACCAAAA, R-TGTAGCACGATTGCAGCATTG), and DZIF N (F-CACATTGGCACCCGCAATC, R-GAGGAACGAGAAGAGGCTTG), and tempRNA 19 for CDC N2 (F-TTACAAACATTGGCCGCAAA, R-GCGCGACATTCCGAAGAA) primers, respectively. Serial dilutions of the tempRNA and primers were tested using 1 µL of a 1E-6 dilution in a 10 µL reaction, corresponding to 699 copies of tempRNA 10, 1890 copies of tempRNA 18, and 699 copies of tempRNA 19. A series of concentrations from 50 to 600 nM in symmetric and asymmetric forward and reverse primer concentrations were added to the RT-qPCR reaction using the GoTaq 1-Step RT-qPCR System (Promega, Madison, WI, USA) according to the manufacturer’s instructions at 60 °C annealing temperature. RT-qPCR was run on a LightCycler480 and analyzed using the corresponding software and LinRegPCR (Version 11.0, Medical Biology, Amsterdam UMC, The Netherlands [25]).

RT-pPCR products were analyzed on agarose gels before excision and purification of DNA fragments of DZIF N product, subjected to analysis by Sanger sequencing using the DZIF N primers.

To obtain absolute reaction efficiencies, 10-fold serial dilutions of template RNA were added to the RT-qPCR reaction (as described above). A semilog fit was performed using GraphPad Prism to determine the slope to determine the RT-qPCR efficiency.

### 2.4. Detergent Lysis Buffer Method Optimization

Lysis buffers of 150 mM NaCl, 10 mM Tris-HCl pH7.5 final concentration were supplemented with final concentrations of a range of concentrations of IGEPAL CA-630, 1 or 10% of Triton X-100, or 5% Tween-20 (all Sigma, Steinheim, Germany) in nuclease-free water, supplemented with RNasin Plus (Promega, Madison, WI, USA) as indicated in individual experiments.

Virus production media (VPM) were generated by inoculating near-confluent Vero E6 cells at MOI 0.1 of EDB-2, EDB-1, or EDB-B117-2 (as determined by endpoint titration on Vero E6 cells—data shown are for EDB-2) for 1.5 h before change of media to complete media and infection for 36 h. Supernatant was harvested and debris cleared by centrifugation at 1500× *g* for 10 min before freezing in aliquots at −80 °C.

For initial optimization, single-round frozen aliquots of VPM were heat-inactivated at 70 °C for 10 min in 100 µL aliquots in thin-walled PCR tubes using a PCR machine to ensure core temperature was reached for the correct amount of time. After validation of SARS-CoV-2 inactivation, experiments were repeated with non-heat-inactivated VPM. No difference in sensitivity, efficiency, or fluorescence strength could be observed.

vRNA was extracted using the QIAamp Viral RNA Mini kit (Qiagen, Hilden, Germany) according to the manufacturer’s instructions and added to the RT-qPCR reaction at equivalent volumes and dilutions corresponding to the amount of VPM added as VPM/lysis buffer to the reaction.

Virus lysis buffer, containing the indicated amounts of detergent and additives, was thoroughly mixed by pipetting at a 1:1 ratio with VPM and incubated for 20 min at room temperature, unless otherwise indicated. Different incubation times were tested and whereas shorter incubation times increased Ct values, no improvement was observed after 20 min.

Unless otherwise indicated, RT-qPCR reactions using the GoTaq 1-Step RT-qPCR reactions were set up in 10 µL final reaction volumes according to the manufacturer’s instructions containing 350 nM CDC N3 primer of each, forward and reverse primer; and VPM/lysis buffer mix was added at 10% final volume. No further reference dye was added. The RT-qPCR was run according to the manufacturer’s instructions on a LightCycler480 using an annealing temperature of 60 °C and analyzed using the corresponding software and LinRegPCR.

### 2.5. Proteinase K Treatment

To test the addition of proteinase K to the lysis buffer, each detergent lysis buffer was supplemented with 0.1 AU/mL proteinase K (Qiagen, Hilden, Germany) and 0.83 mM final concentration EDTA (Sigma, Steinheim, Germany) to prevent heat damage to RNA during heat inactivation. After, mixed 1:1 with VPM or vRNA in nuclease-free water (NF-H_2_O) samples were incubated in thin-walled tubes in a PCR machine for 30 min at 56 °C prior to heat inactivation for 10 min at 95 °C. The inactivated lysate was added at 10% final volume to the RT-qPCR reaction; this was set up and analyzed as described before.

### 2.6. Efficiency Curves

Ten-fold serial dilutions of template RNA or VPM in mock lysate with NF-H_2_O or media were generated as described for the respective experiment. Lysate dilutions were added at 10% final volume to the RT-qPCR reaction, set up and analyzed as described before. A semilog fit was performed using GraphPad Prism to determine the slope for RT-qPCR efficiency.

### 2.7. RNasin and Betaine Addition Experiments

Lysis buffer was supplemented with RNasin Plus as indicated, using a stock concentration of 40,000 U/mL. Lysis and RT-qPCR reactions were performed as described above.

Betaine (Sigma, Steinheim, Germany) was added to the RT-qPCR reaction to the final concentration stated, and RT-qPCR reaction and analysis were performed as described above.

### 2.8. Virus Inactivation

VPM was mixed 1:1 with lysis buffer or PBS and incubated for 20 min at room temperature. A 10-fold serial dilution of lysate was made in inoculation medium, and near-confluent Vero E6 cells inoculated with the diluted virus. Inoculum was removed 1.5 h post inoculation and replaced with culture medium. At 48 hpi, supernatant was harvested and heat-inactivated at 70 °C for 10 min before RNA extraction using the QIAamp Viral RNA Mini kit or lysis of the supernatant before quantification of viral RNA. Due to the toxicity of the lysis buffer mix on cells, half of the lysate/VPM or lysate/PBS mixtures were subjected to a buffer exchange using a 100 kDa MWCO protein concentrator (Pierce, Thermo Scientific, Rockford, IL, USA), reduction to 20% volume and two washes with PBS prior to serial dilution and inoculation of Vero E6 cells as described above.

Viral RNA or lysates were subjected to RT-qPCR analysis using 350 nM of CDC N3 primers in the GoTaq 1-Step RT-qPCR System according to the manufacturer’s instructions and as described above for lysates. If the Ct recorded was 35 or above, a well was classified as non-infected TCID_50_ calculated accordingly.

### 2.9. Testing of Other Enveloped Viruses

vRNA and vDNA were extracted from VPM of IAV and RSV using the QIAamp Viral RNA Mini kit and the DNeasy Blood and Tissue kit (Qiagen, Hilden, Germany) for extraction of BHV-1, respectively, according to the manufacturer’s instructions.

Lysates and vRNA or vDNA were analyzed as described above by RT-qPCR and qPCR, respectively, using the GoTaq 1-Step RT-qPCR and qPCR protocol using a 60 °C annealing temperature. Primers used were: IAV-NA-fwd (ACTGGAAGTCAAAACCATACTGGA) IAV-NA-rev (CCCACGGATGGGACAAAGAG), RSV-L-fwd (GAACTCAGTGTAGGTAGAATGTTTGCA), RSV-L-rev (TTTCAGCTATCATTTTCTCTGCCAAT), BVH-1-CirC-fwd (CCCTGCCGCAAGTTTATGCTGTAT), and BVH-1-CirC-rev (GTAAGAAAGTCGTGCAGTGAATCGG), all used at 400 nM final concentration.

To assess efficiency, serial dilutions of VPM diluted in complete DMEM were subjected to the final protocol for direct lysis RT-qPCR (below), alongside serial dilutions of vRNA or vDNA, respectively.

To assess accuracy, different concentrations of lysate or respective amounts of vRNA or vDNA were added to the (RT-)qPCR reactions.

### 2.10. Recommended Final Protocol for Direct Lysis RT-qPCR

Within appropriate containment facilities and according to local health and safety guidelines (as required for the viral pathogen used), mix equal amounts of virus-containing supernatant with virus lysis buffer (VL buffer—2.5% IGEPAL CA-630, 150 mM NaCl, 1:4000 RNasin Plus, in 10 mM Tris-Cl, pH7.5) by thoroughly pipetting up and down. Addition of a small amount of SDS (to 0.02% final concentration in the lysis buffer) can further improve sensitivity. Incubate the mixture at room temperature for 20 min before decontamination of the containment tube and removal from high containment facilities, according to local health and safety guidelines. Agitation on a plate shaker or equivalent can improve lysis efficiency, especially in small volume containers, such as 96- or 384-well plates.

To minimize pipetting errors, we recommend diluting the lysate in NF-H_2_O before adding to the reaction at larger volumes. If higher accuracy is desired, a lower percentage lysate and, consequently, a higher dilution can be prepared or a lower amount added to the reaction.

## 3. Results

### 3.1. SARS-CoV-2 Primer Optimization Using a 1-Step RT-qPCR Fluorescent Dye System Shows Highest Efficiency and Sensitivity with the CDC N3 Primer

One-step RT-qPCR using fluorescent dye detection is by far the most cost effective method of RT-qPCR; therefore, we focused on optimizing this for use with in vitro applications requiring quantification of SARS-CoV-2 production. Primer pairs N1, N2, and N3 by the US Centers for Disease Control and Prevention (CDC) [26] and the German Center for Infection Research (DZIF) against RdRp and N [27], designed and widely used for the detection of SARS-CoV-2, were selected for optimization in a one-step RT-qPCR reaction using the Promega GoTaq system incorporating the dsDNA-binding dye BYRT green on a LightCycler480.

Increasing concentrations of primer (symmetric and asymmetric concentrations) were tested following the manufacturer’s standard protocol, using target-specific, in vitro transcribed RNA templates of 600–1600 bp length in 10 μL and 20 μL final reaction volumes. An increase in reaction volume to the recommended 20 µL showed no difference in efficacy or sensitivity for any of the primers; therefore, the data shown here illustrate the more cost-effective 10 μL reaction volume.

All primer pairs tested showed clear improvement in sensitivity from 50 to 250 nM of equal primer concentration (Figure 1A,C). Efficiencies, calculated using LinRegPCR, showed a similar increase (Figure 1C). Detected fluorescence was highest for CDC N3, DZIF N, and RdRp (Figure 1A), which can be partly explained by the longer product length of DZIF N (128 bp) and DZIF RdRp (100 bp), but also the higher reaction efficiency, which can be observed by comparing the difference in CDC N primers, all amplifying a roughly 70 bp product (Figure 1C). Using standard conditions with 60 °C annealing temperature, CDC N1, DZIF N and RdRp consistently showed the formation of primer dimer products (Figure 1B). Increasing the concentration of N2 to the recommended concentration of 1000 nM improved its efficiency, but resulted in intermittent primer dimer formation.

Analyzing the bands generated in the PCR reaction, we found that DZIF N formed larger products than the expected 128 bp (Figure 1D). Sanger sequencing was only successful on some of the products but revealed circularization or multiplication of a section of the product. Increasing the annealing temperature did not eliminate the occurrence of these products.

Despite the use of significantly increased primer concentrations for the DZIF N primer, efficiencies did not improve beyond 78.24%. DZIF RdRp showed an efficiency of 89.61% (Figure 1E). However, by far the best performing was primer pair CDC N3 at a symmetrical concentration of 350 nM, showing an efficiency of 93.14% and a sensitivity of <1.89 template copies/reaction (Figure 1E). Paired with the production of primer dimer by CDC N1 and N2, and DZIF RdRp, and the multiple products of DZIF N, CDC N3 at a symmetrical concentration of 350 nM forward/reverse was selected for further development of the method.

### 3.2. IGEPAL-630-Based Buffers Show Highest Efficiency and Sensitivity, Shortly Followed by Triton X-100

As a first step, heat lysis, as described for patient samples, was tested towards release of vRNA from the capsid in virus culture supernatant, hereafter referred to as virus production medium (VPM). However, vRNA release from VPM by heating for 5 min at 95 °C was found to be limited. The difference in Ct values between vRNA extracted using a column RNA purification kit and heat lysis of an equivalent corresponding volume of VPM was found to be >10 cycles, corresponding to a roughly 1000× loss in sensitivity (data not shown). Combined with the requirement for a heat block or, ideally, a PCR machine to ensure correct core temperature during heat inactivation and concerns over RNA stability, heat inactivation was abandoned as a broadly applicable method and focus shifted to a detergent lysis-based method.

Initial optimization experiments were performed using VPM following heat inactivation, at 70 °C for 10 min with a PCR machine to ensure good heat transfer and correct core temperature, to allow processing outside CL3. This virus inactivation method had previously been confirmed by serial dilution and inoculation of cells with heat-inactivated virus (>6log10 TCID_50_/mL reduction). At later stages, once virus inactivation by lysis buffer was confirmed, we validated that there was no difference in Ct values, sensitivity, or efficiency between vRNA extracted from VPM with and without 70 °C heat-inactivation.

All tested lysis buffers were based on a 150 mM NaCl, 10 mM Tris-HCl, pH 7.5 solution supplemented with different lysis detergents for VPM to be lysed at a 1:1 ratio. The salt concentration in the lysis buffer was based on [4], who found a 150 mM NaCl concentration to be most sensitive in the RT-qPCR reaction. VPM, which in most cases is DMEM or RPMI, contains 108–118 mM Cl^-^ and 138–155 mM Na^+^; thus, in a 1:1 dilution with lysis buffer, NaCl concentration will remain just slightly under 150 mM.

Three different detergents were tested to assess their efficacy in releasing vRNA from VPM for RT-qPCR: 1 or 10% Triton X-100, 0.25% IGEPAL-630, or 5% Tween-20 in a buffer supplemented with 10 U/mL RNasin Plus RNA inhibitor. These initial concentrations were chosen either based on available SARS-CoV-2 inactivation data or previous reports in lysis protocols. VPM lysis was performed for 20 min at room temperature. It was also assessed whether proteinase K treatment could improve vRNA release. Therefore, proteinase K was added at 0.1 AU/mL to each detergent lysis buffer, supplemented with 0.83 mM final concentration EDTA to prevent heat damage to RNA during heat inactivation, mixed 1:1 with VPM or vRNA in nuclease-free water (NF-H_2_O), and incubated for 30 min at 56 °C prior to heat inactivation for 10 min at 95 °C. Proteinase K is not inactivated by EDTA and has been shown to be active in high detergent buffers.

The 1% Triton X-100, 5% Tween-20, and 0.25% IGEPAL-630 detergent lysis buffers showed similar sensitivities (Cts 14.75–15.01) compared to purified equivalent amounts of vRNA (Ct 13.53). The qPCR reaction was clearly impaired by the 10% Triton X-100, and 5% Tween-20 buffers, which consistently produced a lower fluorescence. When the detergent lysis buffers were tested in combination with heating with proteinase K for vRNA or in VPM lysis alone, a marked increase by 1.22–1.37 Cts was observed. This increase was even higher for 10% Triton X-100, where proteinase K digest led to a loss of 5.86 Ct. Efficiencies as calculated by LinRegPCR were above 78% using 0.25% IGEPAL-630, 1% Triton X-100, or 5% Tween-20. The 10% Triton X-100, however, shows markedly lower efficiencies (59.2–62.89%) in VPM. All detergents used are likely to work at high efficiencies if used at lower detergent concentrations upon further optimization. We decided to continue optimization of the IGEPAL-630 buffer based on comparable sensitivity at lower concentration to the other detergents (Figure 2A).

### 3.3. Increasing IGEPAL-630 Concentration to 2.5% Improves RT-qPCR Efficacy and Inactivates SARS-CoV-2 Whilst Not Affecting Sensitivity

To improve virus lysis, buffers containing increasing concentrations of 0.25%, 0.5%, 2.5%, and 4% IGEPAL-630 were tested containing 10 U/mL RNasin Plus. VPM was lysed for 20 min at room temperature before adding 1 µL of the 1:1 VPM/lysis buffer to a 10 µL total reaction volume for RT-qPCR analysis. Efficiency decreased slightly with higher IGEPAL-630 concentrations (83.16% at 0.25% to 77.35% at 4% IGEPAL-630). Ct difference compared to an equivalent amount of vRNA was found to be 2.85 cycles in these experiments so further optimizations were necessary to improve sensitivity (Figure 2B).

To test inactivation of SARS-CoV-2 by the lysis, samples were incubated with 0.25 and 2.5% IGEPAL-630 lysis buffer for 20 min. A serial dilution of the lysate or a purified version where the lysis buffer had been exchanged in a 100 kDa protein concentrator with PBS, as well as the VPM without lysis, were inoculated onto a confluent layer of VeroE6 cells. Cell supernatant was replaced 1.5 h post inoculation (hpi) and supernatant was harvested 72 hpi. Produced virus in the supernatant was quantified by RT-qPCR. The 0.25% IGEPAL-630 only reduced SARS-CoV-2 infectivity by >4log10 TCID_50_/mL; however, the 2.5% IGEPAL-630 (at a final concentration of 1.25% in the lysate) showed complete inactivation with a reduction of >6log10 TCID_50_/mL (Table 1).

To test the effect of IGEPAL-630 on the RT-qPCR reaction, a small amount of template RNA (189 copies/reaction) was tested in an RT-qPCR reaction diluted 1:1 in 2.5% IGEPAL-630 buffer or in NF-H_2_O, and added at 1 µL/reaction. No significant decrease in sensitivity could be observed; however, efficiency was decreased both with lower fluorescence and LinRegPCR-calculated efficiency (Figure 2C).

### 3.4. RNases and RT-qPCR Inhibitors in VPM Can Contribute to the Reduced Sensitivity of the Direct Lysis RT-qPCR Reaction

To elucidate why sensitivities in direct lysis VPM RT-qPCR were lower than in extracted vRNA, despite IGEPAL-630 on its own not affecting sensitivity, a series of experiments were performed. A 10-fold dilution series of RNA template were quantified, diluted in either NF-H_2_O or mock lysate (2.5% IGEPAL-630 mixed 1:1 with fresh cell culture media to replicate a lysate). Cts were increased by an average of 8.75 cycles across all concentrations in samples containing media. Only a slight loss of efficiency could be observed in the dilution curve (Figure 2D).

A small amount of RNA template was diluted 1:10 in either NF-H_2_O, 2.5% IGEPAL-630 lysis buffer, fresh cell culture medium or a 1:1 mixture of fresh media and lysis buffer to mimic a lysate, with or without increasing concentrations of RNase inhibitor, incubated for 20 min and tested in an RT-qPCR reaction at 1 µL/reaction (yielding 18.9 copies/reaction). Incubation in 2.5% IGEPAL-630 lysis buffer or fresh media alone led to an increase in Ct of 3.69 or 2.54 cycles, respectively, compared to NF-H_2_O, while incubation in the mixture of both resulted in an increase of 6.29 cycles. This decrease in sensitivity was completely rescued by the addition of as low as 10 U/mL RNasin Plus to the media/2.5% IGEPAL-630 lysis buffer indicating the presence of RNases in both the media and the lysis buffer (Figure 2E).

When the effects of RNase inhibitors on VPM lysate (using 2.5% IGEPAL-630 lysis buffer) were examined, they were not as pronounced as the effects on the mock lysate using fresh cell culture medium. Incubation for 20 min with a high concentration of RNase inhibitors still leads to an increase in Ct of 2.63–2.71 cycles, indicating the presence of RT-qPCR inhibitors in the “spent” but not “fresh” media (Figure 2E).

### 3.5. Neither Betaine Addition nor Increased Primer Concentrations Improve Direct Lysis RT-qPCR Sensitivity

To try to improve direct lysis RT-qPCR efficiency in the face of suspected inhibitors introduced during cell and virus culture, we tested the addition of betaine, known to improve PCR amplification by relaxing secondary structures, and its effect on the direct lysis RT-qPCR reaction. The 2.5% IGEPAL-630 lysis buffer containing 10 U/mL RNasin Plus was tested in combination with increasing amounts of betaine in the RT-qPCR reaction. In addition, 1 M betaine decreased the efficiency of the RT-qPCR significantly, whilst lower concentrations (0.5 M and 0.1 M) showed no decrease in efficiency but a slight decrease in signal. Overall, betaine addition showed no improvement in sensitivity (Figure 2F).

To assess whether the primer concentrations of CDC N3 used were still appropriate for VPM lysate, and whether further improvements in sensitivity and or efficacy could be made, we tested increasing concentrations of primers ranging from 350 to 600 nM in direct lysis RT-qPCR with 2% of the total volume comprising VPM/2.5% IGEPAL-630 lysate with 10 U/mL RNase inhibitor. No asymmetric primer concentrations were tested. Increasing concentrations of primer did not significantly increase sensitivity and only marginally increased efficiency when analyzed using LinRegPCR (Figure 2G). A serial dilution of VPM lysate at 2% lysate per reaction with 350 nM symmetrical forward and reverse CDC N3 primer concentrations showed an efficiency of 106.19%. Increasing primer concentrations were also tested using template RNA incubated in 1:1 in fresh media/2.5% IGEPAL-630 diluted 1:5 in NF-H_2_O to mimic diluted lysate, containing 10 U/mL RNasin Plus (Figure 2G continued).

### 3.6. Lower Percentage of Lysate in the Reaction Increases Sensitivity and Accuracy of Direct Lysis RT-qPCR

In order to test the effect of increased amounts of lysate on the direct lysis RT-qPCR, the addition of 2 or 3 µL of VPM/2.5% IGEPAL-630 lysate to a 10 µL reaction, corresponding to 20 or 30% of the reaction volume, was tested. However, this significantly decreased reaction efficiency and sensitivity (Figure 2H).

Previous results showed that the VPM lysate contained RT-qPCR inhibitors. We therefore tested whether diluting out the lysate in the reaction could improve the accuracy of the quantification and sensitivity compared to equivalent amounts of vRNA. To reduce pipetting errors, VPM lysate was diluted in NF-H_2_O prior to addition to the RT-qPCR reaction. Whilst VPM lysate at 10% reaction volume showed a Ct difference of 1.52 cycles (±0.78 SEM) compared with equivalent vRNA, lowering the concentration of the lysate reduced this further to <1 cycle: 2%, 0.47 cycles (±0.43 SEM); 1%, 0.39 cycles (±0.38 SEM); and 0.5%, 0.46 cycles (±0.38 SEM). Overall, decreasing volumes of VPM lysate in the reaction showed significant improvements in sensitivity and accuracy when compared to equivalent amounts of vRNA; at 2%, 1%, and 0.5% of the total reaction volume, there was no significant difference between vRNA and lysate of equivalent amounts (Figure 2H).

### 3.7. High Sensitivity and Accuracy of Direct Lysis Protocol for RSV and BHV-1 but Less for IAV

Whilst we found the lysis protocol transferrable to other coronaviruses used in our lab, including PRCV, TGEV, hCoV-OC43, and hCoV-229E, we wanted to test broader application to other enveloped viruses. We used VPM of the –ve RNA respiratory syncytia virus (RSV), the segmented –ve RNA influenza A virus (IAV), and the dsDNA bovine herpes virus 1 (BHV-1).

Serial dilutions of VPM in media were lysed in 2.5% IGEPAL-630 lysis buffer for 20 min before assessing by (RT-)qPCR in a 10 µL reaction at 10% 1:1 lysate compared to a serial dilution of vRNA/vDNA. In a second step, decreasing % volumes of lysate in the reaction were tested ranging from 0.5, 1, 2, 5, to 10%.

For RSV, a high reaction efficiency was observed for both the lysate and vRNA RT-qPCR reaction. The 10% reaction volume lysate dilution curve showed a 2.16 ± 0.17 SEM difference over all dilutions and was significantly different from the vRNA results (paired *t*-test, *p* < 0.0001). The detection limit was <0.19 and <0.019 TCID_50_/reaction for virus lysate and vRNA, respectively. When lower % volumes of lysis buffer were used in the reaction, no significant difference between extracted vRNA and VPM lysate were observed (Figure 3A).

For IAV, a low reaction efficiency was observed for both the lysate and vRNA RT-qPCR reaction, indicating further optimization required for the reaction itself. Initial reactions showed a difference of >6 cycles between lysate and vRNA and reduction of % lysate in the reaction showed no improvement (data not shown). Therefore, additional additives were added to the lysis buffer, including EDTA (2 mM), DTT (200 mM), and SDS (0.02%). Whilst all additives improved the lysis, best results were observed with SDS, yielding a 1.83 ± 0.10 SEM cycle reduction (data not shown). The 10% reaction volume lysate dilution curve (including SDS) showed a 4.3 ± 0.24 SEM difference over all dilutions and was significantly different from the vRNA results (paired *t*-test, *p* < 0.0001). The detection limit was <5 and <0.5 pfu/reaction for lysate and vRNA, respectively, which may also be influenced as a whole by a low reaction efficiency. No improvement was observed when lower % volumes of lysate were used in the reaction, indicating lysis of the virion or dissociation of the segmented vRNPs to be limiting factors (Figure 3B).

For BHV-1, a high reaction efficiency was observed for both the lysate and vDNA qPCR reaction. The 10% reaction volume lysate dilution curve showed no significant difference between lysate and vDNA was observed (paired *t*-test, *p* = 0.9398). The detection limit was <0.015 pfu/reaction for both lysate and vRNA. Similarly, no difference could be observed when lower % volumes of lysate were used in the reaction (Figure 3C).

## 4. Discussion

In this study, we show a direct lysis protocol for a one-step RT-qPCR of SARS-CoV-2 in cell culture supernatant, achieving Ct results within one cycle of those obtained using traditional viral RNA purification, whilst inactivating SARS-CoV-2.

The best performing primer pair in our protocols, CDC N3, is designed to recognize all currently known clade 2 and 3 viruses of the *Sarbecovirus* subgenus. It is therefore likely directly applicable to other viruses of this subgenus and, with changes in primers, to other CoVs. Our results once again show the importance of testing for the optimal concentration of PCR primers and checking for primer dimer formation, particularly when using fluorescent dye incorporation RT-qPCR. We found that CDC N1, DZIF RdRp, and to a lower extent DZIF N, show primer dimers at relatively low primer concentrations. We found that DZIF N forms multimers of its product as observed on the agarose gel evaluation and confirmed by Sanger sequencing, which could affect the efficiency and results from probe-based RT-qPCR diagnostic assays.

As mentioned in the introduction, other groups have tested the use of detergents in SARS-CoV-2 patient swabs; using 1% Triton X-100 or 1% Tween-20 increased Ct values of patient samples, which may increase in comparison to RNase inhibitor-treated samples [14], while others found no significant difference in Ct but a decrease in fluorescence intensity/efficiency using Triton X-100 up to 7%/reaction or Tween-20 up to 15%/reaction [9]. This reflects our observation for IGEPAL-630.

This direct lysis protocol offers a good starting point to develop similar protocols for other enveloped virus direct RT-qPCR, since 1% Triton X-100 and 5% Tween-20 show highly promising results with good sensitivity. Modifying the percentage of detergent, especially for Tween-20, will likely improve results. This is particularly interesting since 0.5% Triton X-100, which is the final concentration in a 1:1 lysis buffer/VPM mix, has been shown to completely inactivate SARS-CoV-2 [28]. Tween-20 will need to be used at higher concentrations since live virus can still be recovered from 30 min treatment with 0.5% final concentration Tween-20 [29].

Others found a proteinase K digest prior to heat lysis to be beneficial to detecting SARS-CoV-2 in patient samples [15,16]. In our hands, heat lysis of SARS-CoV-2 VPM gave significantly worse results than using a lysis buffer; therefore, that avenue was not pursued further for in vitro samples. The addition of a proteinase K digest worsened the sensitivity of the RT-qPCR for 0.25% IGEPAL-630, 1% Triton X-100, or 5% Tween-20/VPM and/vRNA. This may be due to the prolonged incubation time, the presence of EDTA, added to protect RNA stability during heat inactivation of the proteinase K, impairing the RT-qPCR reaction, or heat degradation of the RNA.

One of the limiting factors in direct lysis RT-qPCR is that the volume of lysate, and thereby vRNA copies, added to the reaction is limited. However, the direct lysis protocol was found to be sensitive down to <0.0043 TCID_50_/reaction and showing a 1 × 10^6^ dynamic range, which should be more than sufficient for in vitro experiments.

We found that 2.5% IGEPAL-630 lysis buffer concentration, 1.25% final concentration, slightly decreases the reaction efficiency but does not affect the sensitivity of the RT-qPCR. However, reaction sensitivity was strongly affected by fresh cell culture media. The addition of as little as 20 U/mL RNasin rescued the decrease. This is in agreement with the findings of Pearson et al., who also found that adding RNaseOUT RNase inhibitor significantly increased the sensitivity of their direct RT-qPCR of SARS-CoV-2 patient swab samples, achieving Cts only three cycles higher than using RNA purification and RT-qPCR [14]. This suggests that a broad range of RNase inhibitors can be used in direct lysis assays.

With very low amounts of template RNA, 18.9/reaction, it was observed that 2.5% IGEPAL/NF-H_2_O mix showed a decrease of around 3.69 Cts without RNasin, indicating a small contamination with RNases in the lysis buffer and/or small amounts in NF-H_2_O since the addition of RNasin rescued the reduction in a 2.5% IGEPAL/fresh cell culture media mix, even beyond the NF-H_2_O control. It shows that despite careful handling, working in PCR cabinets, and decontamination of surfaces, RNase contaminations can occur easily and may not show at higher concentration RNA. It is therefore further recommended to add RNases to the lysis buffer.

Interestingly, the addition of RNase inhibitors had no impact on the 2.5% IGEPAL/VPM lysis. This is due to the much higher amount of RNA present, an estimated roughly 10,000 more, so the impact of small amounts of RNases may not be visible. Similarly, viral RNA could still be loosely associated with nucleocapsid proteins, protecting its degradation by RNases.

Whilst SARS-CoV-2 overall only contains around 38% of G and C nucleotides combined, the N region is one of the most GC-rich areas of the virus at 47%, with some stretches reaching over 60% GC content. However, the lack of effect of the addition of betaine suggests that there does not seem to be any PCR inhibition by those areas. Whilst previous work shows that betaine concentrations of up to 2 M can improve PCR efficiency [30], we found 1 M betaine to strongly decrease the efficiency of the RT-qPCR reaction. A reduction in PCR efficiency by betaine was observed by [31], who recommend the addition of ethylene glycol or 1,2-propanediol to improve amplification of regions with higher GC contents. We did not pursue this avenue further, since dilution of lysates showed that RT-qPCR inhibitors rather than a lack of RNA accessibility or virion lysis were the cause of decreased sensitivity.

Looking at our combined results, we found that IGEPAL-630 on its own did not reduce RT-qPCR sensitivity. Fresh cell culture media decreased sensitivity, but this was restored by the addition of RNase inhibitors. However, 2.5% IGEPAL-630/VPM lysate was still showing sensitivities 1.52 (±0.78 SEM) Cts lower than extracted vRNA when added at 10% reaction volume. Testing lower percentage reaction volumes shows that the sensitivity difference can be reduced to less than 1 Ct when lysate is diluted out. This indicates that it is not a lack of complete virus lysis but the presence of RT-qPCR inhibitors in the virus infection media that is limiting the sensitivity.

Applying the lysis protocol to other enveloped viruses, we found the direct lysis protocol to show broader applicability to non-segmented, enveloped RNA and DNA viruses. However, further optimization is required for IAV and possibly other segmented RNA viruses more generally. Addition of additives, such as SDS, has improved efficiency in lysis of IAV, as could additional steps, such as heating of lysate. Results for the dsDNA virus BHV-1 show equal results between lysis and DNA extraction and indicate increased cycle numbers in RNA viruses due to lysate in the reaction buffer, inhibiting the reverse transcription step.

## 5. Conclusions

In conclusion, our method allows inactivation and direct RT-qPCR of SARS-CoV-2 in cell culture medium without the need for heat inactivation or proprietary ingredients. This allows for the fast, efficient, and cost-effective analysis of in vitro SARS-CoV-2 experiments, quantifying the entire replication cycle—including release—of the virus. For the highest sensitivity (lowest Ct values), we recommend the use of 10% lysate of the final volume of the RT-qPCR reaction. For the highest accuracy, compared to extracted vRNA, the addition of 1–2% lysate in the final reaction volume is recommended. Since these are low pipetting volumes, a prior dilution step in NF-H_2_O can reduce pipetting errors (Figure 4A).

The direct lysis RT-qPCR protocol allows for the use of clinical samples without the need for recombinant reporter viruses, allowing for quick screening of new variant strains. The protocol also shows broader applicability to other enveloped viruses. It measures the entire replication cycle, including virus release, within a single-step measurement. This increases speed, reduces costs, and improves the ability to identify a broader range of antiviral agents and host genes involved in the replication cycle. Results can be obtained in 2.5–3 h from culture harvest, increasing speed over second round infections or titration assays significantly (Figure 4B).

We have applied this method to a variety of applications so far, including high-throughput screening, assessing the impact of drugs and other inhibitors on SARS-CoV-2 production, quantifying viral titers, both directly and using endpoint titration. The applications in vitro are nearly endless and can be adapted to most virological questions. These first results of us and others show the applicability of this method across the coronavirus family and it is likely that this method can easily be adapted to most other enveloped viruses (Figure 4C).

## Figures and Tables

**Figure 1 viruses-14-00508-f001:**
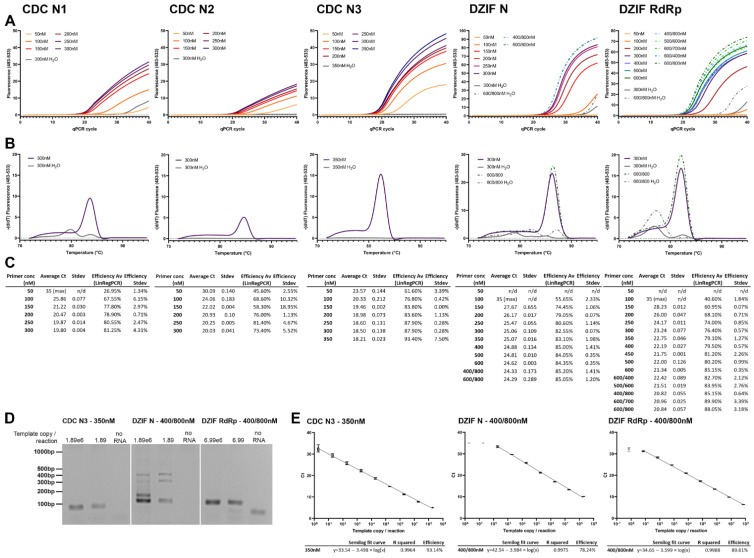
Primer selection for 1-step RT-qPCR amplification of SARS-CoV-2. Increasing symmetric (single number) and asymmetric (forward/reverse separated) concentrations of forward and reverse primer pairs were tested in a 1-step RT-qPCR reaction using templates representing the respective target regions. Primer sets CDC N1, N2, and N3, and DZIF N and RdRp (from left to right) were tested using 699 (CDC N2 and DZIF RdRp) or 1890 (CDC N1 and N3, and DZIF N) template copies per reaction. (**A**) shows the amplification curves of selected primer concentrations. Exact concentrations for each curve are listed in the legend for each graph. Continuous lines represent symmetric concentrations whilst dash/dot curves show asymmetric primer concentrations. Grey curves depict no template reactions. (**B**) shows the melting curves (72–95 °C) of the PCR reaction with the highest shown symmetric, and if shown in (**A**), asymmetric concentration. (**C**) shows the Ct values as calculated using the LightCycler software using the 2nd derivative of the max calculation. Efficiencies for each amplification were calculated using LinRegPCR. *n* = 2; curves represent the average. (**D**) PCR products obtained were analyzed on an agarose gel to assess primer–dimer formation in high and low template (indicated above gel pictures) RT-qPCR reactions. (**E**) Serial dilutions of template were run in RT-qPCR reactions to assess efficiency of the PCR reactions for the best performing primer pairs. A semilog fit curve was calculated using GraphPad prism to assess reaction efficiency from the slope. Samples excluded from the efficiency calculation are greyed out, as they were beyond reaction sensitivity limits. *n* = 3 × 2, error bars represent min and max.

**Figure 2 viruses-14-00508-f002:**
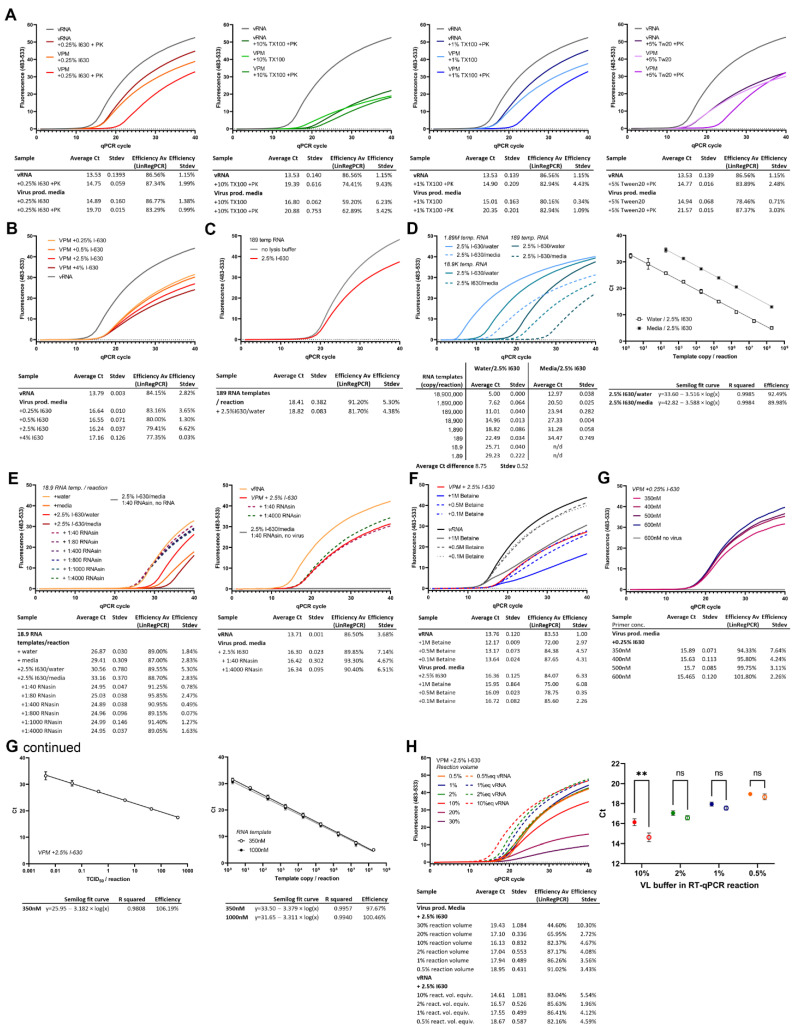
Direct lysis RT-qPCR. Virus production medium (VPM), corresponding amounts of vRNA, or template RNA (as indicated in individual experiments) were lysed 1:1 with detergent lysis buffer for 20 min at room temperature before addition to a 1-step RT-qPCR reaction at 10% of the reaction volume using 350 nM symmetric concentrations of the CDC N3 pair, unless otherwise indicated. Ct values as calculated using the LightCycler software using the 2nd derivative of the max calculation. Efficiencies for each amplification were calculated from amplification curves using LinRegPCR. For serial dilutions, a semilog fit curve was calculated using GraphPad Prism to assess reaction efficiency from the slope. (**A**) Different detergents, 0.25% IGEPAL-630 (I630), 10% and 1% Triton X-100, and 5% Tween 20 (Tw20) were assessed for their ability to release vRNA from cell supernatant and use in a direct lysis RT-qPCR. In parallel to vRNA and a standard lysis of 1:1 VPM:lysis buffer, vRNA and VPM were lysed 1:1 and incubated for 30 min at 56 °C in lysis buffers containing 0.1 AU/mL proteinase K (PK) and 0.83 mM EDTA before heat inactivation for 10 min at 95 °C. *n* = 2. (**B**) Increasing concentrations of IGEPAL-630 were tested in the direct lysis reaction in comparison to equivalent amounts of vRNA in NF-H_2_O. *n* = 2. (**C**) Template RNA in NF-H_2_O was lysed in buffer containing 2.5% IGEPAL-630. *n* = 2. (**D**) A serial dilution of template RNA in NF-H_2_O or fresh cell culture media was incubated in lysis buffer containing 2.5% IGEPAL-630. *n* = 2, error bars represent min and max. (**E**) 18.9 RNA template copies were incubated in lysis buffer containing 2.5% IGEPAL-630 and increasing amounts of RNasin Plus (stock concentration 40 U/µL) or fresh media or NF-H_2_O. Similarly, VPM was incubated using a low or a high amount of RNasin in the lysis buffer, and compared to vRNA extracted from an equivalent amount of VPM. *n* = 2. (**F**) An RT-qPCR reaction was set up containing 10% either VPM lysate using 2.5% IGEPAL-630 or vRNA in NF-H_2_O. Reactions were supplemented with 0.1, 0.5, or 1 M betaine. *n* = 2. (**G**) Increasing symmetric concentrations of CDC N3 primer were added to the VPM/lysate RT-qPCR reaction. *n* = 2. Serial dilutions of VPM in lysis buffer or template RNA in mock lysis buffer were run in an RT-qPCR assay and analyzed as described above. *n* = 3 × 2. (**H**) Decreasing and increasing amounts of VPM lysate were added to the RT-qPCR reaction, corresponding to 0.5–30% of the reaction volume (as indicated for each curve). Corresponding amounts of vRNA in NF-H_2_O were run in parallel. Curves represent an average of 3 × 2, except for 20 and 30%, representing an *n* = 2. Ct values *n* = 3 × 2 show mean ± SEM and were analyzed using a 2-way ANOVA. ** *p* < 0.005 (0.0012).

**Figure 3 viruses-14-00508-f003:**
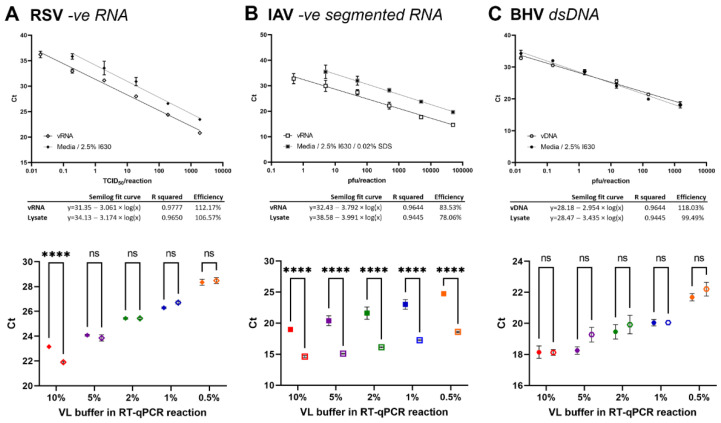
Direct lysis RT-qPCR of other viruses. (**A**) RSV, (**B**) IAV, (**C**) BHV. Top: Serial dilutions of VPM in media lysed in 2.5% IGEPAL-630 lysis buffer and analyzed as described above. *n* = 3 × 2 ± SEM. Bottom: Decreasing amounts of VPM lysate were added to the RT-qPCR reaction, corresponding to 0.5–10% of the reaction volume (as indicated for each curve). Corresponding amounts of vRNA in NF-H2O were run in parallel. *n* = 3 × 2 ± SEM. Analysis using a 2-way ANOVA. **** *p* < 0.0001.

**Figure 4 viruses-14-00508-f004:**
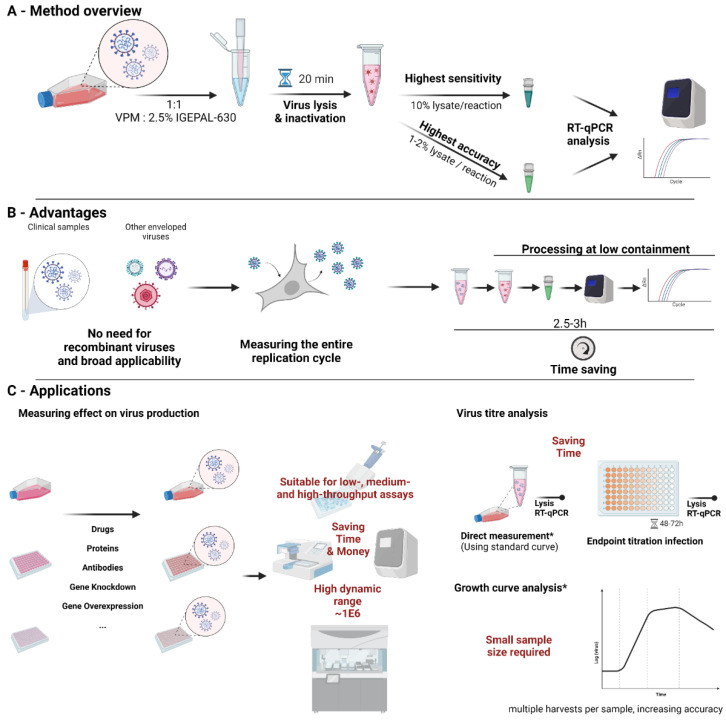
Direct lysis RT-qPCR method. (**A**) depicts an overview for using the method. (**B**) highlights the advantages of using direct lysis RT-qPCR. (**C**) shows applications for which direct lysis was successfully used and suggestions for other applications. * Direct measurement of virus titers may not accurately represent infectious virus dependent on harvesting time and conditions. Similarly, growth curves at later points of infection may not accurately represent infectious virus particles present in the solution. Created with https://biorender.com/ (accessed on 1 January 2022).

**Table 1 viruses-14-00508-t001:** Virus inactivation following a 20 min incubation of virus stock diluted 1:1 with 0.25 or 2.5% IGEPAL-630, or PBS, with or without buffer exchange.

	Mean Virus Titer in log_10_ TCID_50_/mL (95% Confidence Interval)	Titer Reduction in log_10_ TCID_50_/mL (95% Confidence Interval)
PBS 1:1 (neat)	7.05 (6.75–7.22)	-
0.5% I-630 1:1 (neat)	≤2.8 *	≧4.25 (3.95–4.42)
2.5% I-630 1:1 (neat)	≤2.8 *	≧4.25 (3.95–4.42)
PBS 1:1 (buffer exchange)	6.90 (6.34–7.14)	-
0.25% I-630 1:1 (buffer exchange)	2.71 (2.59–2.81)	4.35 (4.05–4.52)
2.5% I-630 1:1 (buffer exchange)	≤0.8 ^♦^	≧6.25 (5.95–6.42)

* Cytotoxicity in undiluted, 1 × 10^−1^, and 1 × 10^−2^ dilutions. Limit of detection for test was 2.8 log10 TCID_50_/mL; 95% confidence interval cannot be calculated. ^♦^ Limit of detection for test was 0.8 log10 TCID_50_/mL; 95% confidence interval cannot be calculated.

## Data Availability

The authors confirm that the data supporting the findings of this study are available within the article. Raw data supporting the findings of this study are available from the corresponding author (C.T.-B.) on request.

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
