# Peer review of "Direct Lysis RT-qPCR of SARS-CoV-2 in Cell Culture Supernatant Allows for Fast and Accurate Quantification"

_viruses, 2022, doi:10.3390/v14030508_

Round 1
Reviewer 1 Report
In this well-written manuscript the authors describe the validation of a useful simplification of in vitro processing of SARS-CoV-2 virus samples as they occur in typical research laboratories.
The study is well designed and has been carefully conducted, providing many useful hints for the improvement of the PCR-based analysis of SARS-CoV-2- and most likely other enveloped viruses.
comment for improvement:
As the study is so straight-forward, it would be nice to see in the same context an application to a second enveloped virus, e.g. influenza, HCV or HIV-1(?) etc. This would allow a broader generalization and applicability of this work.
Author Response
We would like to thank the reviewers and the editor for reviewing our manuscript and for processing our revised version. We think the manuscript has improved and hope that it is acceptable for publication in Viruses in its revised form.
- In this well-written manuscript the authors describe the validation of a useful simplification of in vitro processing of SARS-CoV-2 virus samples as they occur in typical research laboratories.
The study is well designed and has been carefully conducted, providing many useful hints for the improvement of the PCR-based analysis of SARS-CoV-2- and most likely other enveloped viruses.
We thank the reviewer for these positive comments and the appreciation of the work conducted and presented. - comment for improvement:
As the study is so straight-forward, it would be nice to see in the same context an application to a second enveloped virus, e.g. influenza, HCV or HIV-1(?) etc. This would allow a broader generalization and applicability of this work.
Whilst we already found broad applicability of this protocol to other coronaviruses, which we have stated in the revised manuscript, we have taken the reviewers comments on board and have tested applicability of the method to three other enveloped viruses to represent other sizes and genomes. We selected a clinical isolate RSV (-ve RNA), H1N1 IAV PR8 (segmented -ve RNA), and BHV-1 (dsDNA and large). We found great results with RSV and BHV-1 but have seen limitations for IAV, likely due to the vRNP packaging. We have presented these findings in the new figure 3 of the revised manuscript.
Reviewer 2 Report
The study evaluated the direct lysis RT-qPCR method for quantification of SARS-CoV-2 in culture supernatant. It is useful and practical for more researcher advance their studies. It is suggested that the part of Material and Methods is too large and should be simplified. The authors showed the results by graphs and they also suggested to be statistical tests and makes the results more convincing.
Author Response
We would like to thank the reviewers and the editor for reviewing our manuscript and for processing our revised version. We think the manuscript has improved and hope that it is acceptable for publication in Viruses in its revised form.
We take this opportunity to reply to the reviewer's comments.
- The study evaluated the direct lysis RT-qPCR method for quantification of SARS-CoV-2 in culture supernatant. It is useful and practical for more researcher advance their studies. It is suggested that the part of Material and Methods is too large and should be simplified.
We have shortened the Materials and section, particularly "primer optimization and in vitro transcription" and "Recommended final protocol" with smaller changes where appropriate. We hope this is to the reviewer's satisfaction. - The authors showed the results by graphs and they also suggested to be statistical tests and makes the results more convincing.
Statistical tests have been included where appropriate and necessary based on the nature of the results, repeats, and overall conclusions. Standard deviations are included in all tables for readers to draw conclusions. However, many of the data presented add to a composite conclusion.
For example, figure 1, shows primer optimisation and selection, which is based on primer-dimer formation (and risk thereof at higher concentration), product size and efficiency of the reaction. Therefore, the readers are provided with the table results, primer dimer curves, product size, and efficiency calculations - both by linregPCR analysis and dilution curve to draw their own conclusions for the whole picture.
In figure 2, optimisation steps have low repeats, wherefore standard deviations are more appropriate than statistical analysis. Similar to the previous experiments, also curve shape rather than absolute Ct or efficiency are considered towards protocol selection. Where needed and appropriate, for example figure 2H, statistical comparisons have been made and described.
Similarly, new figure 3 has been analysed statistically both in the figure for dilution comparisons and for the dilution curve efficiency as was appropriate and required.